# LaGeM 💎 : A Large Geometry Model for 3D Representation Learning and Diffusion

**Biao Zhang, Peter Wonka**
KAUST, Saudi Arabia
{`biao.zhang, peter.wonka`}@kaust.edu.sa

## Abstract

This paper introduces a novel hierarchical autoencoder that maps 3D models into a highly compressed latent space. The hierarchical autoencoder is specifically designed to tackle the challenges arising from large-scale datasets and generative modeling using diffusion. Different from previous approaches that only work on a regular image or volume grid, our hierarchical autoencoder operates on unordered sets of vectors. Each level of the autoencoder controls different geometric levels of detail. We show that the model can be used to represent a wide range of 3D models while faithfully representing high-resolution geometry details. The training of the new architecture takes 0.70x time and 0.58x memory compared to the baseline. We also explore how the new representation can be used for generative modeling. Specifically, we propose a cascaded diffusion framework where each stage is conditioned on the previous stage. Our design extends existing cascaded designs for image and volume grids to vector sets.

## 1 Introduction

Diffusion models are currently the best-performing models for image, video, and 3D object generation. For 3D object generation, there are two main branches of research. The first branch, pioneered by Dreamfusion (Poole et al., 2022), aims to lift 2D diffusion models to 3D model generation. The advantage of this method is that it can benefit from the large-scale 2D datasets used for training 2D diffusion models and it sparked a lot of follow-up work (Poole et al., 2022; Wang et al., 2023; Lin et al., 2023; Chen et al., 2023; Wang et al., 2024; Qian et al., 2023; Tang et al., 2023; Yi et al., 2023; Wang & Shi, 2023; Liu et al., 2024; Long et al., 2024; Zheng et al., 2024; Li et al., 2023a; Ho et al., 2022; Xu et al., 2023). The second branch tackles the training on 3D datasets directly. The advantage of this method is that it is more direct and leads to faster inference times (Mittal et al., 2022; Yan et al., 2022; Zhang et al., 2022; Zeng et al., 2022; Zheng et al., 2023; Hui et al., 2022; Zhang et al., 2023; Siddiqui et al., 2024; Chen et al., 2024a;b). Our work sets out to contribute to this second branch of methods.

Among these 3D native generation methods, 3DShape2VecSet (Zhang et al., 2023) (or VecSet for short) has been proven to be an effective method to encode 3D geometry. It proposed an autoencoder to find an efficient representation for 3D models as a set of vectors. Because of the high reconstruction quality and compactness of the latent space, the method alleviates the difficulty of training 3D generative models. Some other works (Zhao et al., 2024; Cao et al., 2024; Dong et al., 2024; Petrov et al., 2024; Zhang et al., 2024b; Zhang & Wonka, 2024; Lan et al., 2024b) follow the VecSet representation. We noticed that VecSet's expressiveness is limited by the number of latent vectors. It is overfitting on smaller datasets like ShapeNet and is unable to scale to larger datasets. To improve the expressiveness, we need to scale up the latent size and the training dataset. The straightforward way is to employ hundreds of GPUs for training which is expensive (Zhang et al., 2024b). Thus, our goal is to reduce the training cost in terms of time and memory consumption while achieving similar or even better autoencoding quality.

In the image domain, NVAE (Vahdat & Kautz, 2020) extended the design of the variational autoencoder (VAE) (Kingma, 2013) to a hierarchical VAE based on the design of the U-Net. The latent space of the NVAE is a multi-scale latent grid and the reconstruction quality of the images from the NVAE improves a lot over the VAE. An illustration of the architectures can be found in Fig. 1. We

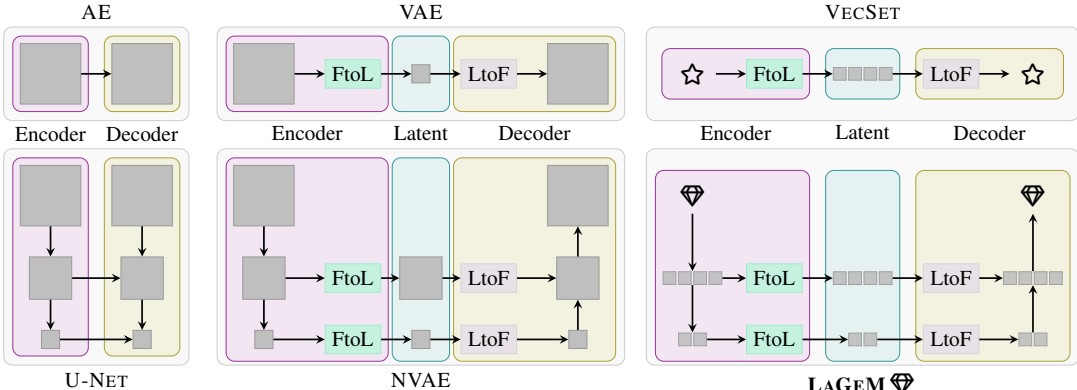

Figure 1: **Autoencoders.** We show different autoencoder architectures here, including AE (AutoEn-coder), U-Net, VAE (Kingma, 2013), NVAE (Vahdat & Kautz, 2020), VecSet (Zhang et al., 2023) and the proposed LaGeM. VAE and NVAE are for image data, while VecSet and LaGeM are for geometry (distance function) data. In the top row, VAE and VecSet use a single-scale latent to represent the data. Both NVAE and LaGeM use multi-scale latents to represent data. All the previous works VAE, NVAE, and VecSet apply KL divergence in the bottleneck to regularize the latent space, while in this work, we apply standardization in the bottleneck.

draw inspiration from the design of the NVAE and design a multi-scale latent VecSet representation, called *LaGeM*. We train our architecture on a large-scale geometry dataset Objaverse (Deitke et al., 2023) and improve training time by 0.7 and memory consumption by 0.58 compared to VecSet.

Additionally, we also propose a cascaded generative model for the hierarchical latent space. We generate the latent VecSet from the lower resolution level to the highest resolution level stage-by-stage. In each stage, we use the previously generated latents as conditioning information. As a result, this enables control over the level of detail of the generated geometry.

| Latents | Controlling |
|---------|-------------|
| Level 3 | Main Structure |
| Level 2 | Major Details |
| Level 1 | Minor Details |

We summarize our contributions as follows:

- We propose a hierarchical autoencoder architecture with faster training time and low memory consumption. The latent space is composed of several levels.
- The model is capable of training on large-scale datasets like objaverse.
- We propose a cascaded diffusion model to generate 3D geometry in the hierarchical latent space. This enables control of the level of detail of the generated model.

## 2    RELATED WORKS

We show an overview of latent 3D generative models in Table 1, particularly focusing on the type of latent space used.

### 2.1    LEARNING METHODS

Usually, a learning method is required to convert 3D geometry to latent space. 1) One way to do this is to convert 3d geometry to latent space with a per-object optimization method, e.g. (Erkoç et al., 2023; Yariv et al., 2024). For larger datasets, this approach is very time-consuming. 2) Alternatively, auto-decoder, e.g., DeepSDF (Park et al., 2019), jointly optimize the latent space for all objects in the dataset. However, as there is no encoder, new objects cannot be mapped to latent space easily. 3) Therefore, a commonly used framework is the auto-encoder. The optimization is efficient because it is performed jointly for all objects in the dataset, and new objects not in the training set can be quickly encoded using the encoder. Thus, we also build on this approach.

Table 1: **Geometric Latent Representation and Generation.**

| Method | Learning Method | Latent Rep | Hierarchies |
|---|---|---|---|
| ShapeFormer (Yan et al., 2022) | AutoEncoder | Sparse Volume | Single |
| 3DILG (Zhang et al., 2022) | AutoEncoder | Irregular Grid | Single |
| LION (Zeng et al., 2022) | AutoEncoder | Latent Points | Multi |
| TriplaneDiffusion (Shue et al., 2023) | AutoDeocder | Planes | Single |
| SDFusion (Cheng et al., 2023) | AutoEncoder | Volume | Single |
| 3DShape2VecSet (Zhang et al., 2023) | AutoEncoder | VecSet | Single |
| HyperDiffusion (Erkoç et al., 2023) | Per-Object Optimization | Network Weight | Single |
| XCube (Ren et al., 2024) | AutoEncoder | Sparse Volume | Multi |
| Mosaic-SDF (Yariv et al., 2024) | Per-Object Optimization | Irregular Grid | Single |
| 3DTopia-XL(Chen et al., 2024c) | Per-Object Optimization | Irregular Grid | Single |
| OctFusion (Xiong et al., 2024) | AutoEncoder | Sparse Volume | Multi |
| LN3Diff (Lan et al., 2024a) | AutoEncoder | Triplanes | Single |
| **LaGeM** ⬦ (Ours) | AutoEncoder | VecSet | Multi |

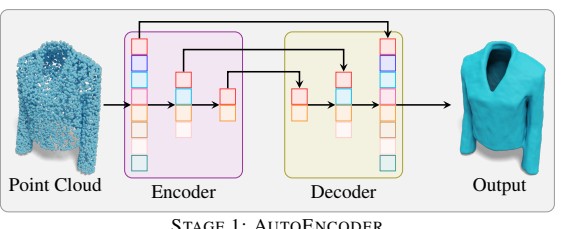 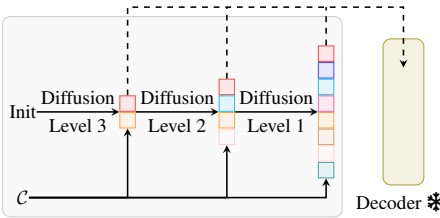

STAGE 1: AUTOENCODER          STAGE 2: LATENT DIFFUSION

Figure 2: **Pipeline.** We proposed a U-Net-style transformer for the autoencoding. In this way, we obtain a hierarchical latent space, which contains several levels of latents. To train the generative diffusion models in the latent space, we propose the cascaded latent diffusion models.

## 2.2 LATENT REPRESENTATIONS

Early methods used regular grids (Yan et al., 2022; Cheng et al., 2023) as the latent representation because of their simple structure. We can easily use convolutional layers to process volume data. To represent high-quality geometric details, we need large-resolution volumes. This makes the training even more difficult because of the $O(n^3)$ complexity. A way to solve this problem is to introduce sparsity (Ren et al., 2024) to the representation like octrees (Xiong et al., 2024) or sparse irregular grids (Zhang et al., 2022; Yariv et al., 2024). Both structures have the potential to represent high-quality 3D models, but generating irregular structures explicitly is difficult for diffusion models. Different from the above-mentioned approaches, 3DShape2VecSet (Zhang et al., 2023) is proposed to solve the reconstruction problem without using any sparse structures. The representation is easy to use. In this paper, we investigate how to improve the VecSet representation. Compared to Zhang et al. (2023), our goal is to obtain an even higher-quality latent space by introducing Level of Latents (LoL).

## 2.3 CASCADED GENERATION

In the field of image generation, there are multiple cascaded diffusion models,e.g., (Ho et al., 2022; Saharia et al., 2022). In the 3D domain, some works (Zeng et al., 2022; Ren et al., 2024) also modeled geometries with hierarchical latents and proposed 3D generative models using cascaded diffusion models. Our work encodes 3D geometry into hierarchical VecSets. Thus, it is straightforward to consider cascaded latent diffusion to train generative models in our latent space.

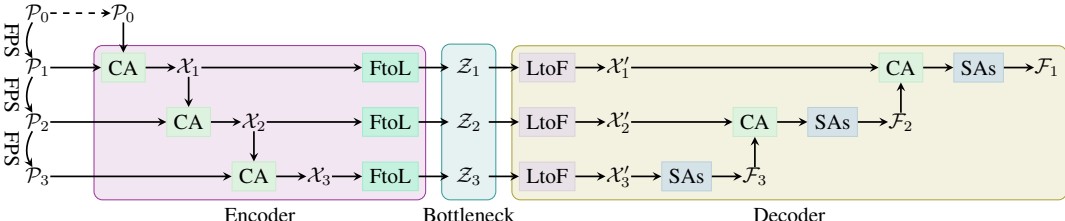

Figure 3: **Geometry Autoencoder.** The design from VecSet (Zhang et al., 2023) can be seen as a special case of the proposed LaGeM network with only one level.

Figure 4: **LaGeM architecture.** We show an illustration with 3 levels of latents.

## 3 METHODOLOGY

### 3.1 BACKGROUND OF VECSET REPRESENTATIONS

The VecSet (Zhang et al., 2023) representation converts a dense point cloud to a latent vector set $\mathcal{Z} = \{\mathbf{z}_1, \mathbf{z}_2, \ldots, \mathbf{z}_M\}$ with $\mathbf{z} \in \mathbb{R}^D$ so that an occupancy/distance function $\mathcal{O}(\mathbf{p})$ can be recovered from the vector set. The simplified network is illustrated in Fig. 3.

**Encoding.** The process first downsamples the 3D input point cloud $\mathcal{P}^{\text{Input}} = \{\mathbf{p}_i\}_{i=1,\ldots,N}$ with furthest point sampling (FPS), $\mathcal{P} = \text{FPS}(\mathcal{P}^{\text{Input}}, r)$, where $r$ is the down-sampling ratio, and $\mathcal{P}$ is a low-resolution version of $\mathcal{P}^{\text{Input}}$. Then $\mathcal{P}^{\text{Input}}$ is converted to an unordered set with cross-attention

$$\text{CA}(Q = \text{PE}(\mathcal{P}), K = \text{PE}(\mathcal{P}^{\text{Input}}), V = \text{PE}(\mathcal{P}^{\text{Input}})) = \mathcal{X} = \{\mathbf{x} \in \mathbb{R}^C\}_{i=1,2,\ldots,M}, \quad (1)$$

where PE is a positional embedding function (Zhang et al., 2023) and $\text{CA}(\cdot, \cdot, \cdot)$ is a cross-attention module. We also write $\text{CA}(\mathcal{P}, \mathcal{P}^{\text{Input}})$ for short. Here, the positional embedding used to project a 3D coordinate $\mathbf{p} \in \mathbb{R}^3$ to the high dimensional space $\mathbb{R}^C$ is omitted for simplicity. To obtain a highly compressed latent space, the vectors in $\mathcal{X}$ are further compressed to a lower-dimensional space $\mathbb{R}^D$ where $D \leq C$ (Feature to Latent, or FtoL in short),

$$\text{FtoL}(\mathcal{X}) = \mathcal{Z} = \{\mathbf{z} \in \mathbb{R}^D\}_{i=1,2,\ldots,M}. \quad (2)$$

This compression step is also regularized by KL divergence.

**Decoding.** Each latent vector in $\mathcal{Z}$ is first converted back to feature space $\mathbb{R}^C$ (Latent to Feature, or LtoF in short),

$$\text{LtoF}(\mathcal{Z}) = \mathcal{X}' = \{\mathbf{x}' \in \mathbb{R}^C\}_{i=1,2,\ldots,M}. \quad (3)$$

The features $\mathcal{X}'$ are fed into a series of self-attention layers to obtain final occupancy/distance function representations $\mathcal{F}$,

$$\text{SAs}(\mathcal{X}') = \mathcal{F} = \{\mathbf{f} \in \mathbb{R}^C\}_{i=1,2,\ldots,M}, \quad (4)$$

where $\text{SAs}(\cdot)$ is implemented using several self-attention layers. Now we can decode a continuous function. For a continuous coordinate in the space $\mathbb{R}^3$, we have

$$\mathcal{O}(\mathbf{p}) = \text{FC}\left(\text{CA}(\mathbf{p}, \mathcal{F})\right) \in \mathbb{R}. \quad (5)$$

See Table 2 for more details on $\text{FtoL}(\cdot)$ and $\text{LtoF}(\cdot)$.

### 3.2 HIERARCHICAL VECSET

The complexity of the self-attention layers in Eq. (4) is $O(M^2)$, i.e., quadratic in the number of input vectors. This severely affects the training time when $M$ is large. However, to represent high-quality geometry details, we usually need a large $M$. This makes training a large VecSet network

Table 2: **Regularization in the Bottleneck.** We compare the proposed regularization (**NBAE**) and VAE. We do not need an explicit loss to regularize the latent space.

| | Features to Latents (FtoL) | | Latent Loss | Latents to Features (LtoF) |
|---|---|---|---|---|
| VAE | $\boldsymbol{\mu} = \mathrm{FC}_\mu(\mathbf{x})$ 
 $\boldsymbol{\sigma} = \mathrm{FC}_\sigma(\mathbf{x})$ | $\mathbf{z} = \boldsymbol{\mu} + \boldsymbol{\sigma} \odot \boldsymbol{\epsilon}$ | KL Divergence | $\mathbf{x}' = \mathrm{FC}_{\mathrm{up}}(\mathbf{z})$ |
| **NBAE** | $\bar{\mathbf{z}} = \mathrm{FC}_{\mathrm{down}}(\mathbf{x})$ | $\mathbf{z} = \dfrac{\bar{\mathbf{z}} - \mathrm{E}[\bar{\mathbf{z}}]}{\sqrt{\mathrm{Var}[\bar{\mathbf{z}}]}}$ | - | $\mathbf{x}' = \mathrm{FC}_{\mathrm{up}}(\mathbf{z} \odot \boldsymbol{\gamma} + \boldsymbol{\beta})$ |

more challenging (for example, $M = 2048$ in CLAY (Zhang et al., 2024a)). Motivated by the design of the U-Net and NVAE (Vahdat & Kautz, 2020), we propose a new network. Specifically, in the design of the U-Net (see an illustration in Fig. 1), image feature grids are downsampled to lower resolutions where some convolution blocks are applied, and then upsampled to the original resolution. In this way, we can avoid performing convolutional layers in high-resolution images (which can be time-consuming). We transferred this idea to the VecSet representations. Two necessary building blocks are operations to down-sample and up-sample a VecSet. Inspired by the design of 3DShape2VecSet (Zhang et al., 2023) (an illustration can be found in Fig. 3), we interpret the cross attention in the encoder part as a down-sampling operator. Similarly, we can also use it for up-sampling. The resulting network is shown in Fig. 4.

We have $L$ levels in the U-Net-style transformer, where we number the levels from one (highest resolution) to $L$ (lowest resolution). For notational convenience, we denote the input point cloud as level 0. In the $i$-th level, we first obtain a lower resolution of the point clouds in the $(i-1)$-th level, $\mathrm{FPS}(\mathcal{P}_{i-1}, r_{i-1}) = \mathcal{P}_i$ where $\mathcal{P}_0$ is the input point cloud. We use cross attention to compress the feature set $\mathrm{CA}(\mathcal{P}_i, \mathcal{P}_{i-1}) = \mathcal{X}_i$. Different from previous approaches, we propose a new to way regularize the latent space,

$$\mathrm{FtoL}(\mathcal{X}_i) = \mathrm{ZeroMeanAndUnitVariance}(\mathrm{FC}_{\mathrm{down}}(\mathcal{X}_i)) = \mathcal{Z}_i, \qquad (6)$$

where we normalize each vector in the set to have zero mean and unit variance $(\mathbf{z} - \mathrm{E}[\mathbf{z}])/\sqrt{\mathrm{Var}[\mathbf{z}]}$ (It is often called *standardization* in machine learning which is used to standardize the features present in the data in a fixed range.). The motivation behind this design is that diffusion starts with Gaussian noise which also has zero mean and unit variance. In this way, we enforce both our latent space and the initial Gaussian noise to have similar properties. To map the latents back to features, we first scale and shift latents back $\mathbf{z} \odot \boldsymbol{\gamma} + \boldsymbol{\beta}$ (both $\boldsymbol{\gamma}$ and $\boldsymbol{\beta}$ are learnable parameters like in Layer Normalization (Lei Ba et al., 2016)),

$$\mathrm{LtoF}(\mathcal{Z}_i) = \mathrm{FC}_{\mathrm{up}}(\mathrm{ScaleAndShift}(\mathcal{Z}_i)) = \mathcal{X}_i'. \qquad (7)$$

Thus we name it Normalized Bottleneck Autoencoder (NBAE). Unlike KL divergence in a VAE, we do not need an explicit loss term for the latent space. See Table 2 for a comparison between the proposed regularization and commonly used KL divergence in VAEs.

Inspired by the down-sampling usage of cross attention in Zhang et al. (2023), we generalize it to *resampling*. Here we use it as *upsampling for unordered set* $\mathcal{F}_i$. Before feeding the features to self-attention layers, we first upsample features $\mathcal{F}_{i+1}$ from lower resolution levels,

$$\mathrm{SAs}(\mathrm{CA}(\mathcal{X}_i', \mathcal{F}_{i+1})) = \mathcal{F}_i. \qquad (8)$$

The query function in Eq. (5) is changed to

$$\mathcal{O}(\mathbf{p}) = \mathrm{FC}\left([\mathrm{CA}(\mathbf{p}, \mathcal{F}_1)| \cdots |\mathrm{CA}(\mathbf{p}, \mathcal{F}_L)]\right) \in \mathbb{R}, \qquad (9)$$

where $[\cdot | \cdot | \cdots | \cdot]$ is the symbol for concatenation. This means we are using features from all levels to build the final (occupancy) function representation.

### 3.3 DIFFUSION

Cascaded Diffusion (Ho et al., 2022) proposed a method for generating high-resolution images. The method is composed of several stages, where each stage is a conditioned diffusion model. Motivated by this, we propose a cascaded latent diffusion model. In Cascaded Diffusion, images generated from the previous stage are used as a condition in the next stage. We build a cascaded latent

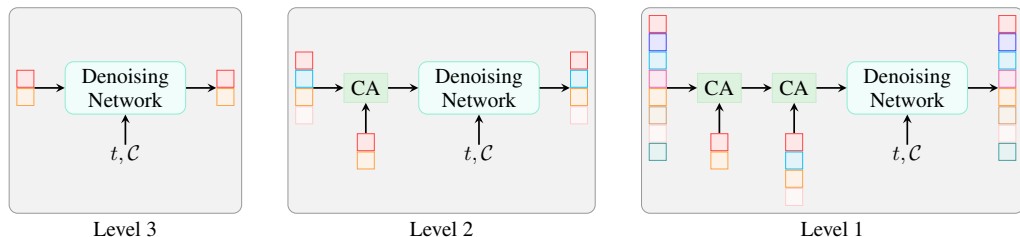

Figure 5: **Cascaded Latent Diffusion.**

Table 3: **Running Statistics of LaGeM.** When using a small number (512) of latent vectors, our model uses 0.87x time and 0.66x memory during training. For larger models (2k latent vectors), the advantage is even more significant (0.7x time and 0.58x memory).

|  | VecSet | LaGeM | VecSet | LaGeM | VecSet | LaGeM |
|---|---|---|---|---|---|---|
| Batch Size | 64 | | 8 | | 4 | |
| Self Attn Layers | 24 | 8/8/8 | 24 | 8/8/8 | 24 | 8/8/8 |
| Attn Channels | 512 | 512/512/512 | 1k | 1k/1k/1k | 1k | 1k/1k/1k |
| # Parameters (M) | 106.13 | 125.15 | 424.24 | 499.85 | 424.24 | 499.85 |
| # Latent Vectors | 512 | 32/128/512 | 2k | 128/512/2k | 2k | 128/512/2k |
| # Latent Channels | 8 | 32/16/8 | 64 | 64/32/16 | 64 | 64/32/16 |
| Training Memory (M) | 56,125 | **37,055** (0.66×) | OOM | **53,791** (-) | 54,543 | **31,662** (0.58×) |
| Training Iteration (sec) | 0.6481 | **0.5658** (0.87×) | - | **0.7714** (-) | 0.6902 | **0.4853** (0.70×) |

diffusion model based on Cascaded Diffusion. Formally, the optimization goal (for our three-level implementation) is as follows,

$$\min_{D_3} \left\| D_3(\tilde{\mathcal{Z}}_3(t), t, \mathcal{C} \qquad) - \mathcal{Z}_3 \right\|,$$
$$\min_{D_2} \left\| D_2(\tilde{\mathcal{Z}}_2(t), t, \mathcal{C}, \mathcal{Z}_3 \quad) - \mathcal{Z}_2 \right\|, \qquad (10)$$
$$\min_{D_1} \left\| D_1(\tilde{\mathcal{Z}}_1(t), t, \mathcal{C}, \mathcal{Z}_3, \mathcal{Z}_2) - \mathcal{Z}_1 \right\|,$$

where $D_i$ is a denoising network, $t$ represents timestep or noise level, $\tilde{\mathcal{Z}}_i(t)$ is the noised version (at timestep $t$) of the latent, $\mathcal{C}$ is optional condition information (*e.g.*, text, images, or categories). The network design is based on DiT (Peebles & Xie, 2022). To generate latents $\mathcal{Z}_i$, we need latents from previous stages $\mathcal{Z}_{>i}$. For diffusion-based image super-resolution methods, this is often done by bilinearly interpolating small images and concatenating them with denoising networks' inputs. As shown in the previous section, we use cross-attention for resampling (both down-sampling and upsampling). Here, we also utilize cross-attention to upsample a latent set. Specifically, assuming we are training a denoising network for $\mathcal{Z}_2$, the input of the network is $\tilde{\mathcal{Z}}_2(t)$,

$$\text{CA}(\tilde{\mathcal{Z}}_2(t), \mathcal{Z}_3). \qquad (11)$$

Similarly, for $\mathcal{Z}_1$,

$$\text{CA}(\text{CA}(\tilde{\mathcal{Z}}_1(t), \mathcal{Z}_3), \mathcal{Z}_2). \qquad (12)$$

In this way, we are gathering information from previous stages. See Fig. 5 for an illustration of the pipeline.

## 4 EXPERIMENTS

### 4.1 AUTOENCODING MODEL

The main autoencoding experiment is trained on Objaverse (Deitke et al., 2023). Models are zero-centered and normalized into the unit sphere. Since most 3D models in this dataset are not watertight,

Table 4: **Evaluation on ShapeNet.** We compare our results to VecSet (Zhang et al., 2023) trained on ShapeNet. If we train our model on ShapeNet and evaluate on ShapeNet our model is slightly better than VecSet. When our model is trained on Objaverse and evaluated on ShapeNet, we can see a very large improvement. Note that it is difficult to scale VecSet to Objaverse training.

| | Chamfer ↓ ($\times 100$) | | | | F-Score ↑ ($\times 100$) | | | |
| | VS | LaGeM($\Delta$) | | | VS | LaGeM($\Delta$) | | |
| | | ShapeNet | | Objaverse | | ShapeNet | | Objaverse |
|---|---|---|---|---|---|---|---|---|
| table | 2.46 | 2.48 | 0.02 | **2.09** | -0.37 | 99.94 | 99.97 | 0.02 | **99.96** | 0.02 |
| car | 5.99 | 5.89 | -0.10 | **4.36** | -1.63 | 89.85 | 90.31 | 0.46 | **92.15** | 2.30 |
| chair | 2.92 | 2.89 | -0.03 | **2.01** | -0.91 | 96.40 | 96.49 | 0.09 | **99.91** | 3.51 |
| airplane | 1.78 | 1.81 | 0.03 | **1.58** | -0.21 | 99.50 | 99.48 | -0.02 | **99.78** | 0.29 |
| sofa | 2.64 | 2.63 | -0.01 | **2.25** | -0.39 | 98.92 | 99.04 | 0.11 | **99.60** | 0.67 |
| rifle | 1.78 | 1.77 | -0.01 | **1.44** | -0.34 | 99.88 | 99.88 | -0.01 | **99.94** | 0.06 |
| lamp | 4.36 | 4.44 | 0.08 | **2.37** | -2.00 | 96.78 | 97.18 | 0.39 | **99.43** | 2.64 |
| mean (selected) | 3.13 | 3.13 | 0.00 | **2.30** | -0.83 | 97.33 | 97.48 | 0.15 | **98.68** | 1.36 |
| mean (all) | 4.68 | 4.63 | -0.04 | **2.42** | -2.26 | 93.25 | 93.47 | 0.23 | **98.93** | 5.68 |

Table 5: **Generalization on Various Datasets.** Our trained model is capable of doing inference on several existing datasets. It can be applied on non-watertight datasets like ABO and pix3d even if the model is trained on watertight datasets. Note that models from ShapeNet were not originally watertight. We use the watertight version processed by (Zhang et al., 2022).

| Dataset | # Meshes | Manifold | Chamfer ↓ ($\times 100$) | | | F-Score ↑ ($\times 100$) | | |
| | | | VS | LaGeM($\Delta$) | | VS | LaGeM($\Delta$) | |
|---|---|---|---|---|---|---|---|---|
| Thingi10k (Zhou & Jacobson, 2016) | 10k | Yes | 4.52 | **2.99** | -1.53 | 92.75 | **97.19** | 4.44 |
| ABO (Collins et al., 2022) | 8k | No | 4.91 | **3.66** | -1.26 | 92.52 | **94.91** | 2.39 |
| ShapeNet (Chang et al., 2015)-test | 2k | Yes | 3.25 | **2.33** | -0.92 | 97.41 | **99.49** | 2.08 |
| EGAD (Morrison et al., 2020) | 2k | Yes | 3.27 | **2.82** | -0.45 | 99.02 | **99.76** | 0.74 |
| GSO (Downs et al., 2022) | 1k | Yes | 3.78 | **2.35** | -1.43 | 94.70 | **99.54** | 4.84 |
| pix3d (Sun et al., 2018) | 700 | No | 6.53 | **6.02** | -0.50 | 87.25 | **87.96** | 0.71 |
| FAUST (Bogo et al., 2014) | 100 | Yes | 2.10 | **1.31** | -0.79 | 99.58 | **99.90** | 0.32 |

we use ManifoldPlus (Huang et al., 2020) to make all meshes watertight. Due to failures of modeling loading and conversion, we obtained around 600k watertight models for training. The three levels of latents are $128 \times 64$, $512 \times 32$, and $2048 \times 16$ (where 64, 32, and 16 are channels of the latents). Some other hyperparameters of the network can also be found in Table 3. We name the model as LaGeM-Objaverse. We also apply the method to ShapeNet, where the train split is taken from (Zhang et al., 2022). Since ShapeNet is a relatively small and easy dataset compared to Objaverse, we choose smaller latents which are $32 \times 32$, $128 \times 16$, and $512 \times 8$. The model is named as LaGeM-ShapeNet. Both models are compared against VecSet (Zhang et al., 2023). We use Chamfer distance and F-score as the metrics. The results are shown in Table 4. Like (Zhang et al., 2023), we first compare the results on the largest categories (which have several thousand training samples) in ShapeNet and then all categories. We can see that, LaGeM-ShapeNet has almost the same number of parameters as VecSet, but with much shorter training time and less training memory. The quantitative results (averaged over all ShapeNet categories) are also better than VecSet's. While for LaGeM-Objaverse, there is a large improvement in both training cost and quantitative results. The quantitative results show an improvement of almost 50 percent averaged across the complete dataset in terms of the metric Chamfer. This demonstrates that LaGeM-Objaverse has good generalization ability. This can also be seen in Fig. 6. The results of LaGeM-Objaverse are good on small categories of ShapeNet. In previous works (Zhang et al., 2023), this is nearly impossible because of limited training samples.

To further prove the generalization ability of LaGeM-Objaverse, we also test the autoencoding on various datasets, including Thingi10k (Zhou & Jacobson, 2016), ABO (Collins et al., 2022),

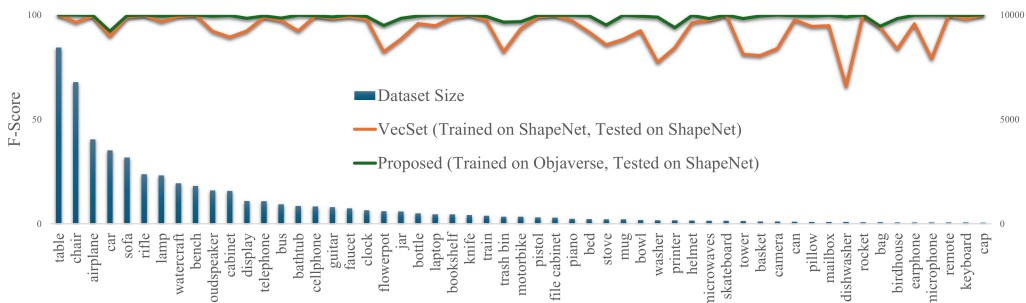

Figure 6: **Generalization on ShapeNet.** Our results are better than VecSet in all categories. On small categories, the results of VecSet are not stable because of limited training samples. In contrast, our trained model also performs well in these categories.

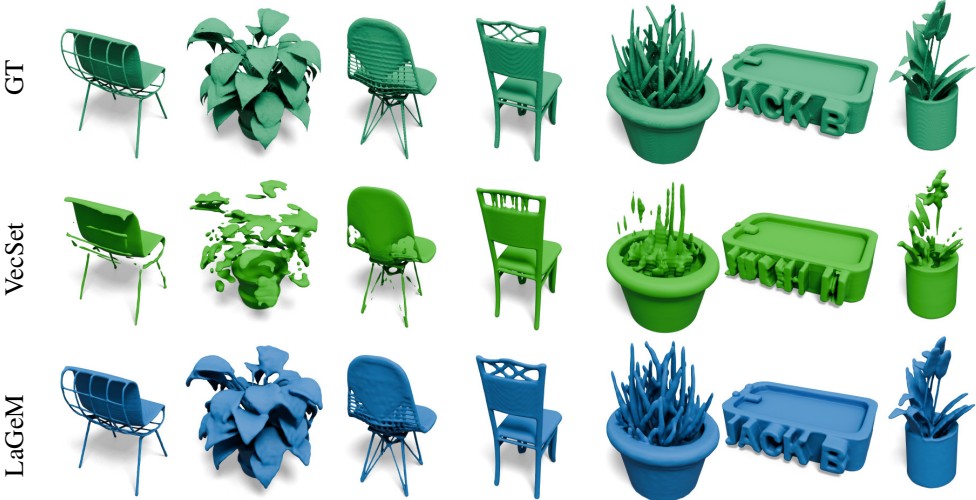

Figure 7: **Qualitative Results on ShapeNet.** We show autoencoding results on ShapeNet. We use VecSet as the baseline. Our model is capable of reconstructing detailed geometry, especially thin structures.

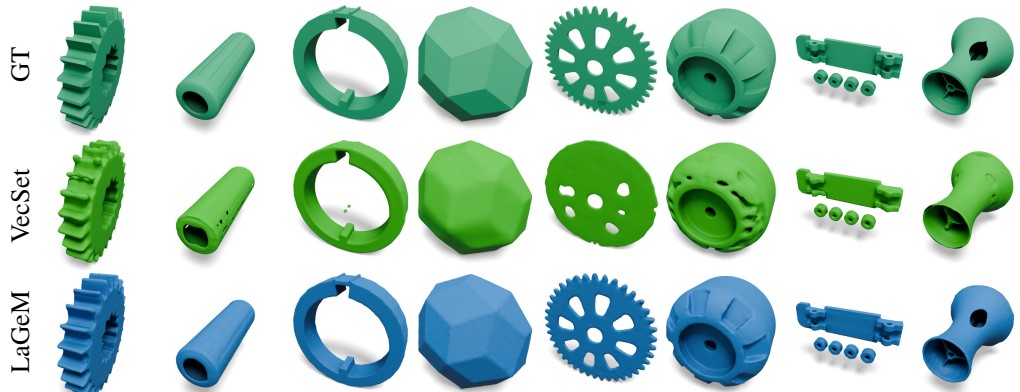

Figure 8: **Qualitative Results on Thingi10k.** Our model can even preserve highly detailed geometry in CAD models.

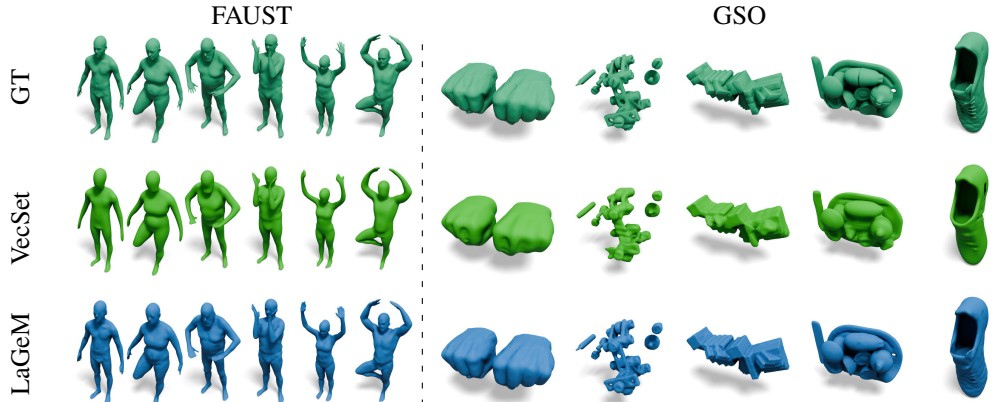

Figure 9: **Qualitative Results on FAUST and GSO.** Results of VecSet are over-smoothed, while our method can preserve sharp details.

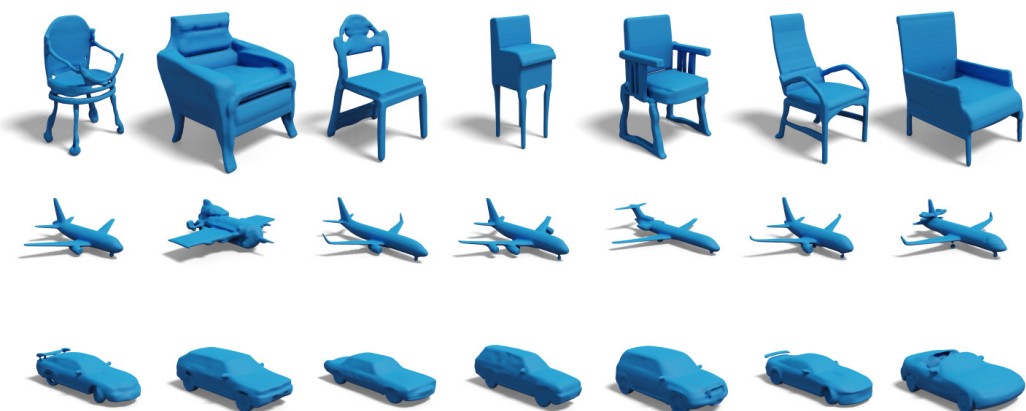

Figure 10: **Category-Conditioned Generative Results on ShapeNet.**

EGAD (Morrison et al., 2020), GSO (Downs et al., 2022), pix3d (Sun et al., 2018) and FAUST (Bogo et al., 2014). The objects from these datasets vary from daily objects, CAD models, human models, and synthetic objects. The quantitative results can be found in Table 5. Again, we use VecSet's model as the baseline. From the metrics, we can see that LaGeM-Objaverse is able to represent different kinds of objects with highly detailed geometry and sharp features. Note that, even for non-watertight meshes, the model is still able to do reconstruction. Visual results of the method can be found in Fig. 7, Fig. 8, Fig. 9.

## 4.2 GENERATIVE MODEL

We conducted two generative experiments; one was on ShapeNet with categories as the condition, and the other one was an unconditional generation on Objaverse-10k. For ShapeNet, the denoising networks of the 3 levels have 12 self-attention blocks with 768 channels. We trained the model for around 200 hours with 4 A100 GPUs. The results are shown in Fig. 10. For Objaverse-10k, due to limited training GPU resources, we select a subset of 10k models from Objaverse and train the unconditional generative model. There are 24 self-attention blocks with 768 channels in all stages of the latents. The model is trained on 16 A100 GPUs for around 100 hours. See Fig. 11 for some unconditional generation results.

**Controllability of the Latents.** We verify that different levels of latents control different levels of detail of the generated samples. During generation, we first generate higher-level latents $\mathcal{Z}_3$, which determine the main structures of the 3D models. Then we use $\mathcal{Z}_3$ as a condition to generate $\mathcal{Z}_2$, which adds major details to the models. In the end, we generate $\mathcal{Z}_1$ conditioned on both $\mathcal{Z}_3$ and $\mathcal{Z}_2$. This final step adds some minor details to the samples. A visual illustration can be found in Fig. 12.

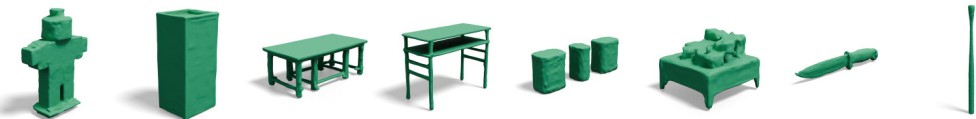

Figure 11: **Unconditional Generative Results on Objaverse-10k.**

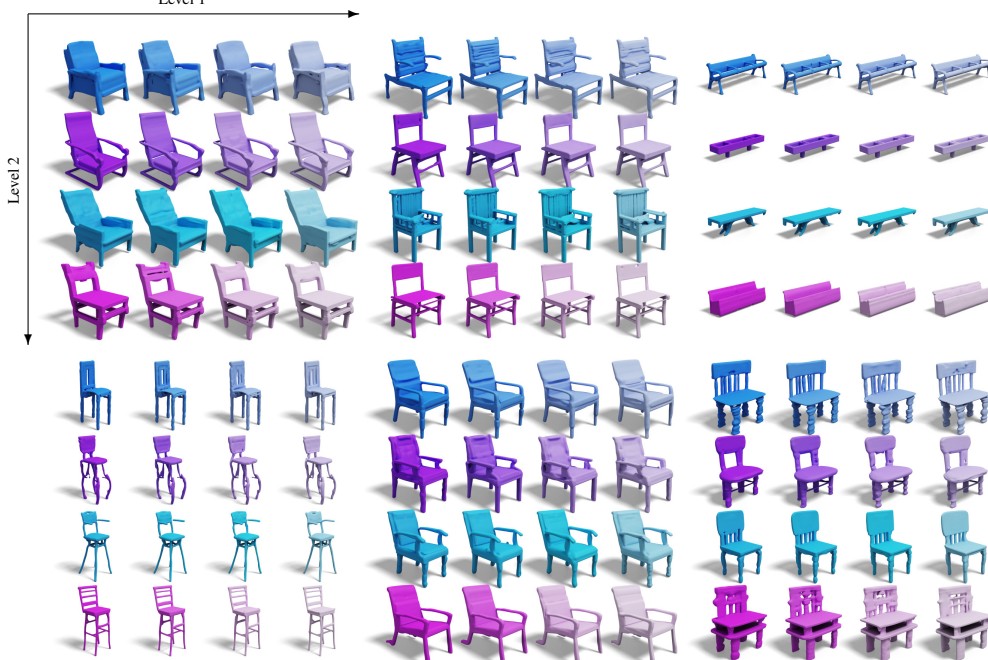

Figure 12: **Latent Levels.** Each small $4 \times 4$ block shares the same level 3 latents $\mathcal{Z}_3$. 3D models in the same block have similar structures. In each block, every $1 \times 4$ line shares the same level 2 latents $\mathcal{Z}_2$. In each line of a block, 3D models look almost the same except for some minor details. Thus, we argue that $\mathcal{Z}_3$ controls the *structure*, $\mathcal{Z}_2$ affects the *major details* and $\mathcal{Z}_1$ is responsible for *minor details*.

## 5 CONCLUSION

We proposed LaGeM (Large Geometry Model), an architecture for encoding 3D geometry. Different from previous approaches, the latent space is modeled as a hierarchical latent VecSets. To make this work, our model employs a U-Net-style design and a new regularization technique for the bottleneck. We showed that this model can be trained much faster with much lower GPU memory costs, especially for larger networks and datasets. This enables scaling of the network for large-scale datasets. We release our model trained on a 600k geometry dataset. Additionally, we proposed a cascaded diffusion model to show some preliminary generative results with the hierarchical latent space.

**Limitation.** Since the latent space is divided into multiple levels, training a diffusion model on all levels still takes a lot of time. Our method does not solve the high training cost problem of diffusion itself.

### ACKNOWLEDGMENTS

The work is supported by funding from KAUST - Center of Excellence for Generative AI, under award number 5940, and the NTGC-AI program.

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

# A   DATA PREPROCESSING

The data preprocessing is based on (Zhang et al., 2022).

## A.1   VOLUME POINTS SAMPLING.

We sample volume points uniformly in the bounding sphere.

```
N_vol = 250000
vol_points = np.random.randn(N_vol, 3)
vol_points = vol_points / np.linalg.norm(vol_points, axis=1)[:, None] *
    np.sqrt(3)
vol_points = vol_points * np.power(np.random.rand(N_vol), 1./3)[:, None]
```

## A.2   NEAR POINTS SAMPLING

The near-surface points are obtained by sampling Gaussian-jittered surface points.

```
N_near = 125000
# surface_points: N_near x 3
near_points = [
    surface_points + np.random.normal(scale=0.005, size=(N_near, 3)),
    surface_points + np.random.normal(scale=0.05, size=(N_near, 3)),
]
near_points = np.concatenate(near_points)
```

## B  DATA AUGMENTATIONS

**Random axis scaling.**   The augmentation is from (Zhang et al., 2022). We randomly sample a scaling factor for each axis which ranges from [0.75, 1.25].

**Unit sphere normalization.**   We normalize each mesh to a unit sphere, i.e., the max point norm of the point clouds is 1.

```
# v: vertices n x 3
v = v - (v.max(axis=0) + v.min(axis=0)) / 2
distances = np.linalg.norm(v, axis=1)
scale = 1 / np.max(distances)
v *= scale
```

**Random rotations.**   We apply random rotations during the training of the autoencoder,

$$\mathbf{R}(\alpha, \beta, \gamma) = \begin{bmatrix} \cos\alpha & -\sin\alpha & 0 \\ \sin\alpha & \cos\alpha & 0 \\ 0 & 0 & 1 \end{bmatrix} \begin{bmatrix} \cos\beta & 0 & \sin\beta \\ 0 & 1 & 0 \\ -\sin\beta & 0 & \cos\beta \end{bmatrix} \begin{bmatrix} 1 & 0 & 0 \\ 0 & \cos\gamma & -\sin\gamma \\ 0 & \sin\gamma & \cos\gamma \end{bmatrix}, \tag{13}$$

where $\alpha$, $\beta$, and $\gamma$ are yaw, pitch, and roll, respectively. Our meshes are firstly normalized into a unit sphere. Thus after the random rotations, the models will still be inside of a unit sphere.

## C  REGULARIZATION

The proposed regularization (see Table 2) is implemented with layer normalization (PyTorch code).

```
# network definition
self.ftl_proj = nn.Linear(x_dim, z_dim)
self.ftl_norm = nn.LayerNorm(dims, elementwise_affine=False, eps=1e-6)
# network forward
z = self.ftl_norm(self.ftl_proj(x))
```

## D  TRAINING TIME QUERY POINTS SAMPLING

In the previous work (Zhang et al., 2022), the sampling strategy is uniformly sampling 1024 points in the bounding volume during training. We found this is not working on Objaverse. Since lots of meshes have very thin structures, this strategy will cause no inside points to be sampled during training. This heavily imbalanced data classification severely affects the occupancy loss.

We propose the following solution. In each iteration, we make sure half of the points have positive labels and the other half have negative labels.

## E  TRAINING LOSS

The loss is binary cross entropy as in previous work (Zhang et al., 2022). Formally, we have

$$\mathcal{L} = \mathbb{E}_{\mathbf{p} \in \mathbb{R}^3} \left[ \text{BCE} \left( \hat{\mathcal{O}}(\mathbf{p}), \mathcal{O}(\mathbf{p}) \right) \right]. \tag{14}$$

In practice, we use the empirical loss

$$\mathbb{E}_{\mathbf{p} \in \mathcal{Q}^{\text{vol}}} \left[ \text{BCE} \left( \hat{\mathcal{O}}(\mathbf{p}), \mathcal{O}(\mathbf{p}) \right) \right] + 0.1 \cdot \mathbb{E}_{\mathbf{p} \in \mathcal{Q}^{\text{near}}} \left[ \text{BCE} \left( \hat{\mathcal{O}}(\mathbf{p}), \mathcal{O}(\mathbf{p}) \right) \right]. \tag{15}$$

Here, $\mathcal{Q}^{\text{vol}}$ is the set of volume query points, and $\mathcal{Q}^{\text{near}}$ is the set of near-surface query points.

## F  DIFFUSION

We use the formulation EDM (Karras et al., 2022) for the diffusion models. The inference/sampling algorithm is also taken from this paper.

$\mathcal{Z}_3, \mathcal{Z}_2, \mathcal{Z}_1$     $\mathcal{Z}_3, \mathcal{Z}_2, \mathcal{Z}_1$     $\mathcal{Z}_3, \mathcal{Z}_2, \mathcal{Z}_1$     $\mathcal{Z}_3, \mathcal{Z}_2, \mathcal{Z}_1$

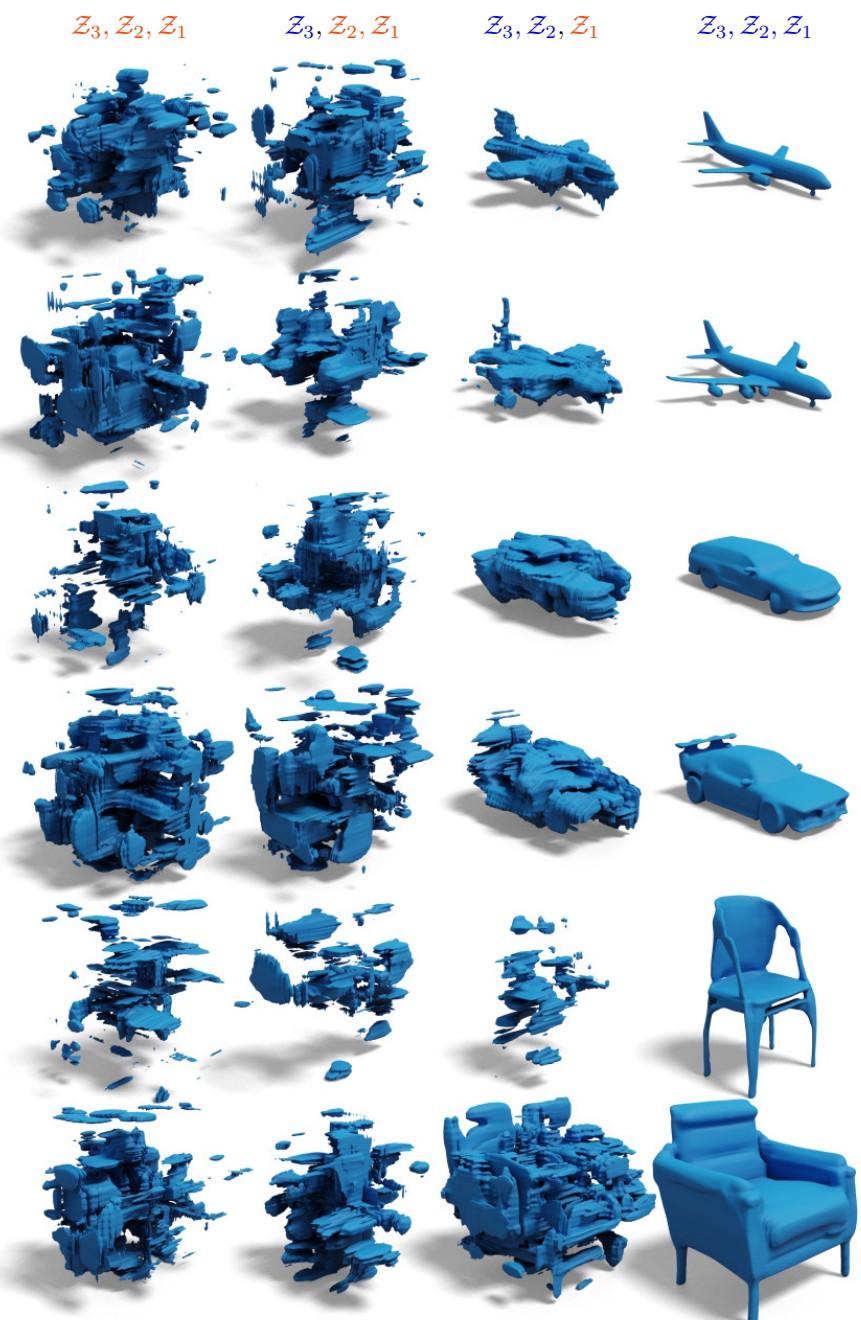

Figure 13: Latent with red color $\mathcal{Z}$ means it is replaced by Gaussian noise. Latent with blue color $\mathcal{Z}$ means it is generated with the diffusion models.

## G   LATENTS ANALYSIS

We analyze how latents are affecting the final reconstruction. The latents are partially replaced by standard Gaussian noise (this is because our latents are also zero mean and unit variance). We show the visual results in Fig. 13.

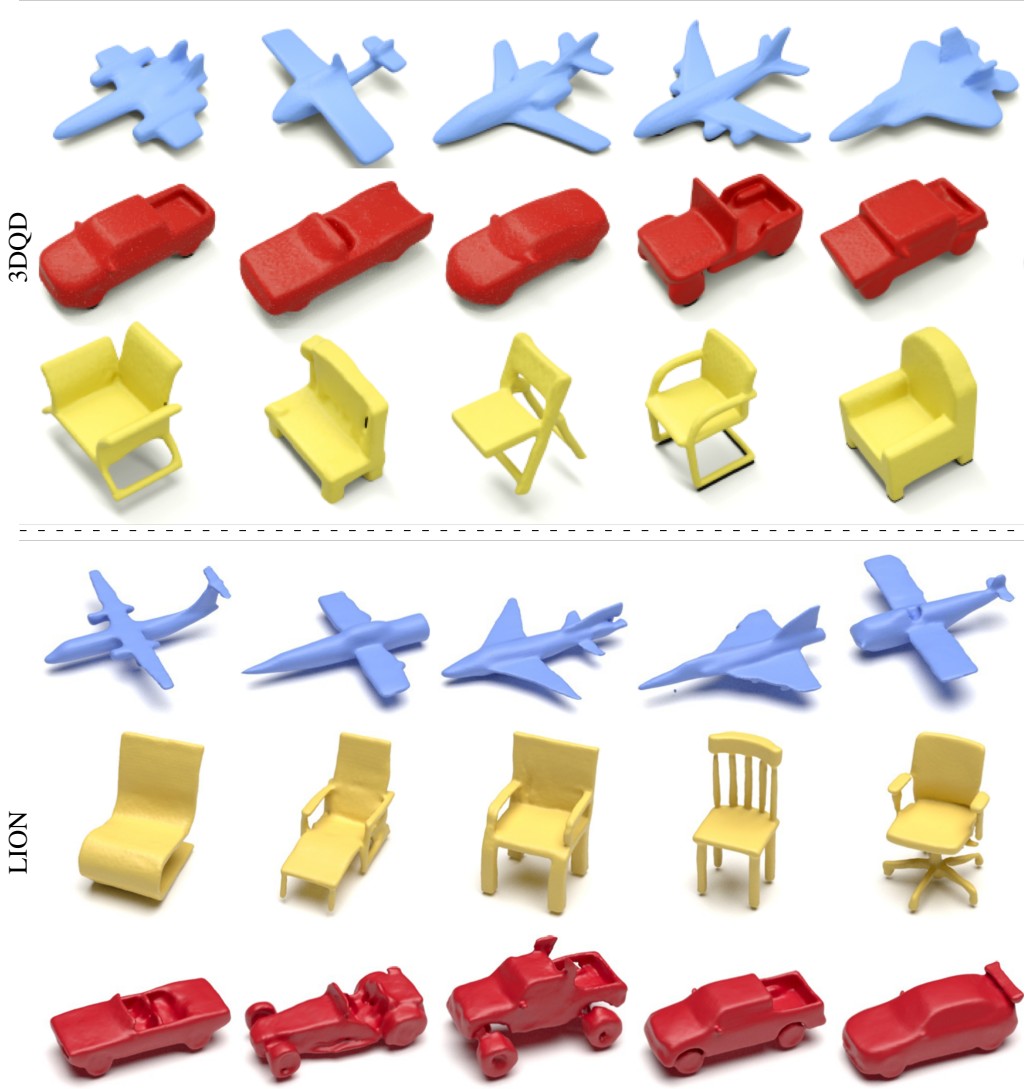

Figure 14: Results from 3DQD (Li et al., 2023b) and LION (Zeng et al., 2022).

Table 6: Generative result comparison.

| chair | 3DILG | VecSet | Ours | table | 3DILG | VecSet | Ours |
|---|---|---|---|---|---|---|---|
| surface-FPD | 0.96 | 0.76 | **0.64** | surface-FPD | 2.10 | 1.19 | **1.12** |
| surface-KPD ($\times 10^3$) | 1.21 | 0.70 | **0.57** | surface-KPD ($\times 10^3$) | 3.84 | 1.87 | **1.72** |

## H  MORE COMPARISONS

We show additional visual comparisons with 3DQD and LION in Fig. 14.

## I  ADDITIONAL GENERATIVE METRICS

We present results in Table 6 for the "chair" and "table" categories for a diffusion model using the generative metrics proposed in VecSet. Both models are trained on ShapeNet and have similar numbers of parameters, as detailed in Table 3 of the main paper.

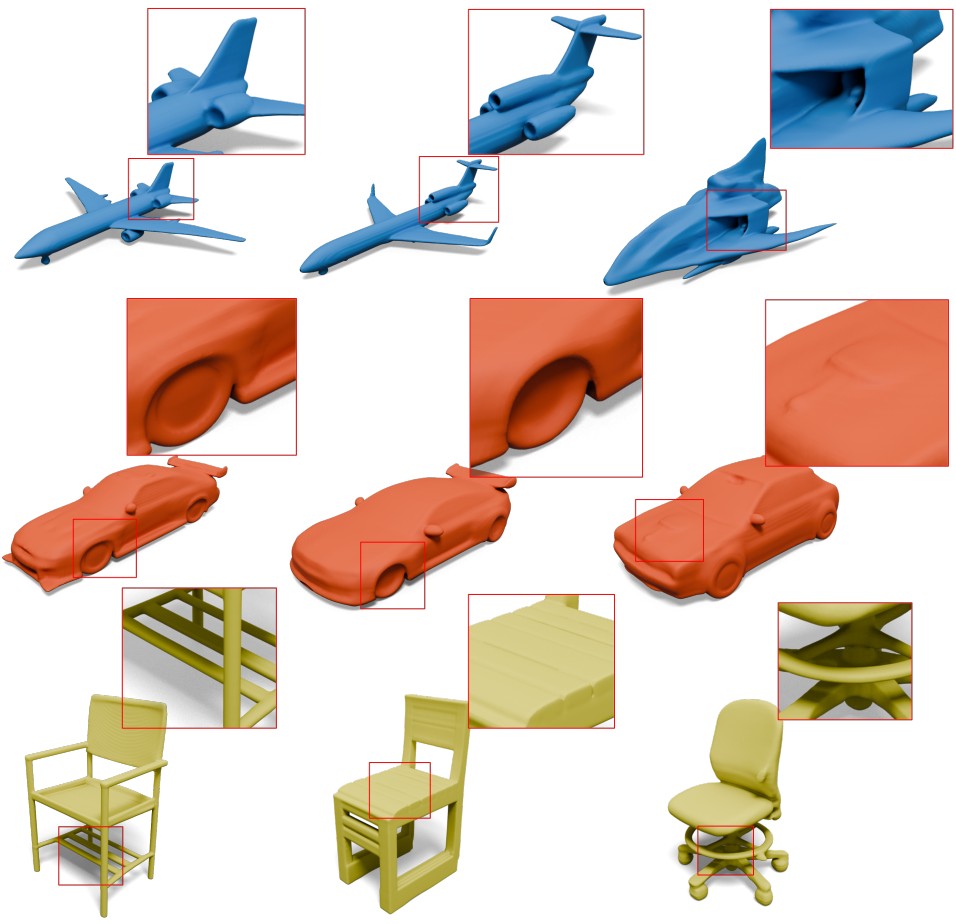

Figure 15: Our generated results. Compared to Fig. 14, we can generate clean, sharp, and detailed shapes.

Table 7: Results on different regularizers.

| Loss | 16 epochs | 32 epochs | 48 epochs | 60 epochs | 72 epochs |
|------|-----------|-----------|-----------|-----------|-----------|
| KL   | 0.1438    | 0.0444    | 0.0276    | 0.0217    | 0.0183    |
| Ours | 0.1320    | 0.0409    | 0.0271    | 0.0217    | 0.0183    |

## J    ABLATION STUDY FOR THE NEW REGULARIZAR

We can also show that our regularizer has similar reconstruction behavior in Table 7. To evaluate the new regularizer, we conducted an ablation study using the VecSet codebase. We compared the volume reconstruction loss of both methods and observed that the performance is nearly identical, with the new regularizer even showing an advantage during the initial epochs. This demonstrates that the proposed regularizer can achieve results comparable to the commonly used KL divergence.

