# OpenReview forum: "LaGeM: A Large Geometry Model for 3D Representation Learning and Diffusion"
_ICLR.cc/2025/Conference — ICLR 2025 Poster_

### Official Review · Reviewer_J6bH · 2024-10-22

**Soundness:** 2
**Presentation:** 3
**Contribution:** 3
**Rating:** 8
**Confidence:** 4

**Summary:**

This paper introduces a novel hierarchical autoencoder that maps 3D models into a highly compressed latent space. The hierarchical autoencoder is specifically designed to tackle the challenges arising from large-scale datasets and generative modeling using diffusion. The paper also proposes a cascaded diffusion framework where each stage is conditioned on the previous stage based on the above hierarchical autoencoder.

**Strengths:**

1. The paper is clearly written and effectively expresses the motivation and role of hierarchical learning.

2. The reporting of experimental hyperparameters is very detailed, enhancing the technical solidity of the paper.

3. The model has been tested on a large number of datasets, validating the robustness of the proposed method.

4. Additionally, I enjoy Fig. 13.

**Weaknesses:**

The main weakness of this paper lies in the lack of comparison experiments.

1. Regarding reconstruction accuracy, this work is only compared with one baseline (VecSet). In reality, there are more comparative options for reconstruction, including autoSDF[1], LION[2], and 3DQD[3].

2. Additionally, this work does not compare the quality of generated results with other baselines. The authors have listed several potential comparison baselines in Table 1. From the perspective of visualizing generated quality, the quality of the generated results does not show a significant advantage over many baselines listed in Table 1.

3. There is also a lack of ablation experiments, such as the rationale behind using 3 levels of latent features.

[1] AutoSDF: Shape Priors for 3D Completion, Reconstruction and Generation

[2] LION: Latent Point Diffusion Models for 3D Shape Generation

[3] 3DQD: Generalized Deep 3D Shape Prior via Part-Discretized Diffusion Process

**Questions:**

The authors claim that it is difficult to scale VecSet to Objaverse training (#L328) and why?

**Details Of Ethics Concerns:**

No ethics review is needed.

---

> ### Author Response · Authors · 2024-11-21
> **response to Reviewer J6bH**
>
> ## Comparisons.
>
> We have added more visual comparisons in Section H of the Supplemental material, including results for LION and 3DQD. The visual outcomes demonstrate our ability to generate clean, sharp, and detailed shapes. Also, note that our results are from a category-conditioned generation on ShapeNet while 3DQD trains each category separately. We will discuss more in the next revision.
>
> ## Why 3 levels.
>
> Conducting such an ablation study was challenging due to our limited GPU resources. During the development of our architecture, we initially tried 2 and 3 levels and found 3 levels to perform better. We have not yet experimented with 4 or 5 levels. We can add this ablation study in an eventual final version of the paper.
> ## Why is scaling VecSet to Objaverse difficult?
>
> We observed that scaling VecSet to more complex datasets beyond ShapeNet requires increasing the number of latents. However, due to the quadratic complexity of self-attention—the core component of VecSet—this demands substantial GPU resources. For instance, CLAY (SIGGRAPH 2024) utilized 2048 latents and 256 GPUs for training. Therefore, our goal is to explore alternative approaches to reduce the training cost. We will emphasize it in the next revision.

---

> > ### Comment · Reviewer_J6bH · 2024-11-25
> >
> > I would like to express my appreciation for the efforts you have made. However, I noticed that you mentioned "Appendix H" in your revisions. Could you please guide me on where I can locate the section?

---

> > > ### Author Response · Authors · 2024-11-25
> > >
> > > Thanks for the reply. It seems the PDF has not been updated due to some mistakes. We just updated the PDF. It is on page 17 and 18.

---

> > > > ### Comment · Reviewer_J6bH · 2024-11-26
> > > >
> > > > I've read the author's response carefully, and I will raise my rating.

---

### Official Review · Reviewer_Uog4 · 2024-11-03

**Soundness:** 3
**Presentation:** 3
**Contribution:** 3
**Rating:** 8
**Confidence:** 5

**Summary:**

This paper introduces a novel 3D autoencoder called LAGEM, which maps 3D models into a highly compressed latent space. The key contribution of this paper is the hierarchical autoencoder architecture, which takes 0.70x the time and 0.58x the memory compared to the baseline. Experiments on Objaverse and ShapeNet demonstrate promising results.

**Strengths:**

1. The paper introduces a novel hierarchical 3D autoencoder with faster training time and low memory consumption, addressing the expressiveness of SoTA method 3DShape2VecSet which is unable to scale to larger datasets.
2. Extensive experiments demonstrate the method's efficacy, outperforming previous state-of-the-art methods on key datasets.
3. The paper is generally well-written and easy to follow. The figures are helpful in illustrating hierarchical architecture.

**Weaknesses:**

1. My main concern is whether having a better 3D autoencoder will lead to a better 3D generative model. On one hand, a better 3D autoencoder implies a higher upper bounder on the quality of the generated results. On the other hand, it requires that the latent space be smoother and easier to learn. Therefore, it would be even better if quantitative metrics for the 3D generated results could be provided.
2. Training a diffusion model on multi-levels takes a lot of training time. So is it possible to only train a single diffusion model to generate latent codes. And train a feed-forward network which take the latent code from the previous level as input to predict the latent code for the next level.
3. Is it possible to generalize the LaGeM to scene-level datasets like Matterport3D or the Replica dataset?

**Questions:**

Please refer to the weaknesses above.

---

> ### Author Response · Authors · 2024-11-21
> **response to Reviewer Uog4**
>
> ## Generative models metrics.
>
> We present results for the "chair" and “table” category for a diffusion model using the generative metrics proposed in VecSet. Both models are trained on ShapeNet and have similar numbers of parameters, as detailed in Table 3 of the main paper.
> | chair                | 3DILG | VecSet | Ours |
> |----------------------|-------|--------|------|
> | surface-FPD          | 0.96  | 0.76   | __0.64__ |
> | surface-KPD (x 10^3) | 1.21  | 0.70   | __0.57__ |
>
>
> | table                | 3DILG | VecSet | Ours |
> |----------------------|-------|--------|------|
> | surface-FPD          | 2.10  | 1.19   | __1.12__ |
> | surface-KPD (x 10^3) | 3.84  | 1.87   | __1.72__ |
>
> ## Diffusion on multi-levels.
> We agree that training multiple diffusion models can be expensive. In the past, we did experiment with both alternatives, training a cascade of diffusion models and training mixed diffusion and reconstruction models as suggested by the reviewer. Both of these approaches are compatible with our framework, but we did not want to focus on this particular aspect in the current submission, since it is somewhat orthogonal to our main contribution. It would definitely be an interesting exploration for future work.
> ## Scene datasets.
>
> Yes, it is possible to generalize LaGeM to scene-level datasets like Matterport3D or the Replica dataset, but it would require several considerations. Scene-level datasets typically involve more complex structures and larger-scale environments compared to simpler object-level datasets. The challenges might be:
> 1. Scene-level datasets contain more intricate details, requiring the model to handle larger and more diverse latent spaces. This might involve adjusting the number of latents, and increasing the model's capacity to capture finer details.
> 2. The number of training samples in Matterport3D or Replica is small, making overfitting quite likely. This might be the main obstacle to train on scene level datasets.

---

> > ### Comment · Reviewer_Uog4 · 2024-11-27
> >
> > I have read the rebuttal carefully and would like to thank the authors. I greatly appreciate the authors for addressing some of my concerns regarding the quantitative metrics for 3D generation. I would like to maintain my original ranking.

---

### Official Review · Reviewer_uqmz · 2024-11-04

**Soundness:** 2
**Presentation:** 3
**Contribution:** 2
**Rating:** 6
**Confidence:** 4

**Summary:**

The paper combines the idea of hierarchical VAE (Vahdat & Kautz, 2020) with VecSet (Zhang et al., 2023), leading to a 3D shape auto-encoder with hierarchical latent space. The proposed mode takes point clouds as input, and produces an occupancy field, which follows VecSet. It is more efficient than a single-level VecSet autoencoder, while being scaleable to large 3D datasets. The proposed auto-encoder outperforms VecSet in terms of reconstruction quality, while being more efficient. When coupled with a cascaded generative model (e.g. a cascaded diffusion model), this also enables control over the individual level of detail of the generated shapes.

**Strengths:**

* The proposed autoencoder significantly outperforms VecSet, the previous work, in terms of reconstruction quality and generalization capability.
* The paper demonstrates the controllability of individual level-of-detail, which is not possible with previous works due to having only a single level of latent vectors.

**Weaknesses:**

* It is well-known that having KL divergence on the latent trades reconstruction quality for a smoother and more compact latent space. Such a well-behaved latent space could be helpful for the performance of the upstream generative model. I thus won't consider removing the KL loss "new regularization technique" (L477-478).
* The idea of controllability over different level-of-detail is interesting. However, according to Figure 13, its effect is very subtle and not particularly useful. This does increase the complexity and training cost of the diffusion model however, as stated in the Limitation section.
* The effect of using different latent regularizations (Table 2) is heavily discussed in Sec. 3.2 but not ablated.
* Table 3: It would be clearer if each column had a header explaining their differences and giving the setting a name.

**Questions:**

For the experiments presented in Table 4, Table 5 and Figure 8, I wonder if both LaGeM and VecSet are using the same latent size and type of latent regularization? If not, the comparison could be unfair, as these differences could dominate the performance difference rather than the hierarchical architecture itself.

---

> ### Author Response · Authors · 2024-11-21
> **response to Reviewer uqmz**
>
> ## KL-regularizer vs. our new regularizer.
>
> First, we would like to clarify that we do not only eliminate the KL-divergence, but we replace it by a specific normalization. Our reason why this is an important contribution pretty much follows the argument of the reviewer. The KL-divergence has proven to be important, because it regularizes the latent space in a way that is beneficial for the downstream generative model. Therefore, it has been widely used. However, we propose an alternative that has shown to be useful in reconstruction as well as generation. Our regularizer is mainly interesting, because it works with the downstream generative model. In addition, our regularizer has another important practical benefit. Training an effective autoencoder requires carefully balancing the weights between the reconstruction loss and the KL loss, which is time-intensive. In our case, with multiple levels of latents, this would involve tuning three separate KL weight terms, further increasing the resource demands to find the optimal model. Our regularizer is much easier to tune.
>
> ## Controllability and complexity.
> We agree that the levels of detail are not as intuitive as in the image domain. In the image domain, a cascaded diffusion model (e.g, Imagen [1]) produces outputs of different resolution. Such a simple progression is not observable in our visualizations. However, we think this work is unique, because it is 3D and it is a cascaded latent diffusion model. We are not aware of other cascaded diffusion models that operate in latent space. We believe that our initial results are interesting and unique, but it may require training separate encoder or regularizer to better extract different levels of detail in future work.
>
> However, we disagree with the implication that the levels of detail would be the only benefit of the cascaded model. Without the cascaded model it would require a lot more GPU resources to train the autoencoder or the diffusion model. That is the main benefit of the cascaded model we introduce.
>
> ## Ablation study on new regularizar
> We can also show that our regularizer has similar reconstruction behavior below. To evaluate the new regularizer, we conducted an ablation study using the VecSet codebase. We compared the volume reconstruction loss of both methods and observed that the performance is nearly identical, with the new regularizer even showing an advantage during the initial epochs. This demonstrates that the proposed regularizer can achieve results comparable to the commonly used KL divergence.
> | Loss | 16 epochs | 32 epochs | 48 epochs | 60 epochs | 72 epochs |
> |------|-----------|-----------|-----------|-----------|-----------|
> | KL   | 0.1438    | 0.0444    | 0.0276    | 0.0217    | 0.0183    |
> | Ours | 0.1320    | 0.0409    | 0.0271    | 0.0217    | 0.0183    |
>
> ## Clear name.
>
> We will update the descriptions to make them clear.
>
> ## Fair comparison
>
> We present results for the "chair" and “table” category using the metrics proposed in VecSet. Both category conditioned generative models are trained on ShapeNet and have similar numbers of parameters (we believe this is the fair comparison).
> | chair                | 3DILG | VecSet | Ours |
> |----------------------|-------|--------|------|
> | surface-FPD          | 0.96  | 0.76   | __0.64__ |
> | surface-KPD (x 10^3) | 1.21  | 0.70   | __0.57__ |
>
>
> | table                | 3DILG | VecSet | Ours |
> |----------------------|-------|--------|------|
> | surface-FPD          | 2.10  | 1.19   | __1.12__ |
> | surface-KPD (x 10^3) | 3.84  | 1.87   | __1.72__ |
>
> [1] Photorealistic Text-to-Image Diffusion Models with Deep Language Understanding

---

> ### Author Response · Authors · 2024-11-26
>
> We hope our response has addressed your questions. As the discussion phase approaches its conclusion, we would greatly appreciate your feedback and would like to know if you have any remaining concerns we can address. Thank you once again for your time and effort in reviewing our work.

---

> > ### Comment · Reviewer_uqmz · 2024-11-26
> >
> > I would like to thank the authors for the response and the category-conditioned generation experiment. I want to follow up on the "Fair comparison" question. I wonder what are the latent embedding counts and dimensions used for this comparison for both LaGeM and VecSet?

---

> > > ### Author Response · Authors · 2024-11-27
> > >
> > > Thanks for the reply.
> > >
> > > In the question "fair comparison", the size of the latents are as follows (length x channels):
> > >
> > > VecSet: 512 x 32 (=16384)
> > >
> > > LaGeM: 512 x 8, 128 x 16, 32 x 32 (=7168)

---

> > > > ### Comment · Reviewer_uqmz · 2024-12-01
> > > >
> > > > Thanks for the clarification! I raised my rating as all my concerns have been addressed.

---

### Official Review · Reviewer_g1aY · 2024-11-04

**Soundness:** 3
**Presentation:** 3
**Contribution:** 3
**Rating:** 6
**Confidence:** 4

**Summary:**

This paper introduces a novel hierarchical autoencoder that compresses 3D models into a highly compact latent space, designed to handle large-scale datasets and support generative modeling using diffusion. Unlike previous approaches, the hierarchical autoencoder works on unordered sets of vectors, with each level controlling different geometric details. The model effectively represents a wide range of 3D models while preserving high-resolution details, and it reduces training time and memory usage compared to the baseline. Additionally, the authors propose a cascaded diffusion framework for generative modeling in the hierarchical latent space, allowing control over the level of detail in generated 3D models.

**Strengths:**

- The proposed method extends prior work VecSet to a hierarchical architecture, which improves generalization ability.

- The hierarchical autoencoder encodes the 3D shape into different levels of latent representations, with each level controlling different geometric details. This feature is highly beneficial for 3D generation.

- The writing in this paper is clean and easy to follow. The comparison with previous work (Table 1) is a valuable addition.

**Weaknesses:**

- The improvement in training time (by 0.7×) and memory consumption (by 0.58×) does not seem significant. In this case, having three levels of latent representations might be too heavy.

- The experiments presented in the paper involve up to 2K latent representations, which is not a substantial sequence length for Transformers with Flash Attention. Recent work [a] has scaled up to 64K latents.

- When using three levels of latent representations, we need three levels of diffusion models, which may lead to additional error accumulation. It would be worthwhile to mention how this issue is addressed for the proposed diffusion model.

- For the proposed regularization, it seems to force the datasets to share the same mean and standard deviation (Eq. 7), which could negatively impact model performance. In contrast, for KLD, very small values are typically used to avoid harming reconstruction. One suggestion is to try the method from [b] without applying any regularization.

[a] Meshtron: High-Fidelity, Artist-Like 3D Mesh Generation at Scale. https://openreview.net/forum?id=mhzDv7UAMu

[b] AutoDecoding Latent 3D Diffusion Models. NeurIPS 2023.

**Questions:**

For the rebuttal, please refer to the Weaknesses section. Additionally, I have a few questions and suggestions:

- For the diffusion transformer, conducting cross-attention only once may not be sufficient.

- In Figure 1, there is no explanation for 'FtoL'.

---

> ### Author Response · Authors · 2024-11-21
> **response to Reviewer g1aY**
>
> ## Improvement.
>
> We consider this a significant improvement over the baseline, particularly for those with limited access to large GPU resources. If this would be a small model then it can be easy to train in any case. However, in our case this makes the difference between being able to train and not being able to train with limited GPU resources.
> ## Flash Attention
>
> This is an orthogonal contribution to our work. We can also use FlashAttention in our implementation to obtain an improvement. However, switching to FlashAttention creates an unfair comparison to previous work VecSet.
> Interestingly, the work [a] also employs a UNet-style-design to reduce the training complexity.
>
> ## Cascaded diffusion
>
> This approach is well-established in the image domain. For instance, both [1] and [2] employ cascaded image generative models. Early large-resolution image diffusion models faced resource constraints, making it challenging to train a single model capable of directly generating high-resolution images. Consequently, hierarchical structures proved beneficial. We believe a similar case applies to the 3D domain. We do not think our framework has a unique type of error accumulation that would require separate handling.
>
> ## Regularization
>
> The new regularization is equivalent to the LayerNorm used in transformers, and in practice, we implement it using LayerNorm (by setting elementwise_affine false as shown in the supplemental).  Our results indicate that it does not negatively impact performance, but only brings advantages.
> Most importantly, KL divergence requires difficult weight tuning. Finding the optimal weight, however, requires extensive tuning and significant GPU resources. This is even more difficult for a three level cascaded model. The proposed regularizer avoids this need for weight tuning, offering a more resource-efficient solution.
> For generation, we show that it produces a high quality latent space in the paper.
> For reconstruction, we conducted an experiment using the VecSet codebase, replacing the KL loss with the new regularization while keeping everything else unchanged. The results showed similar performance.
>
> | Loss | 16 epochs | 32 epochs | 48 epochs | 60 epochs | 72 epochs |
> |------|-----------|-----------|-----------|-----------|-----------|
> | KL   | 0.1438    | 0.0444    | 0.0276    | 0.0217    | 0.0183    |
> | Ours | 0.1320    | 0.0409    | 0.0271    | 0.0217    | 0.0183    |
>
> Regarding [b], as suggested, we reviewed the method. It is an auto-decoder-based approach, which we have discussed in our related works. However, it does not apply to autoencoders. We will discuss this properly in the next revision.
>
> ## Cross-attention in diffusion transformers
>
> The internal structure of the diffusion transformers remains unchanged. For conditional information injection, cross-attention is applied in every block.
>
> ## Figure 1 caption.
>
> We will fix it in the next revision.

---

> > ### Author Response · Authors · 2024-11-21
> > **references**
> >
> > [1] Photorealistic Text-to-Image Diffusion Models with Deep Language Understanding
> >
> > [2] Cascaded Diffusion Models for High Fidelity Image Generation

---

> > ### Comment · Reviewer_g1aY · 2024-11-26
> >
> > Hi, thanks for your rebuttal. I would like to keep my rating.

---

### Meta-Review · Area_Chair_XRWw · 2024-12-14

**Metareview:**

The paper introduces a hierarchical autoencoder for 3D shapes that allows to train diffusion models in for generation in latent space.
The paper was well-received by all reviewers, converging to positive scores, recommending acceptance. The reviewers highlighted the good writing and extensive experiments, clearly showing benefits over previous methods. After rebuttal, remaining concerns regarding ablations and missing metrics have been resolved.

I agree with the reviewers and follow their suggestion with an accept recommendation.

**Additional Comments On Reviewer Discussion:**

The paper already received positively leaning reviews in the first round. Main criticisms were missing ablations and evaluated metrics. The authors provided these in the answer, leading to two reviewers increasing their score.

---

### Decision · Program_Chairs · 2025-01-22

Accept (Poster)